# Tackling Heavy-Tailed Rewards in Reinforcement Learning with Function Approximation: Minimax Optimal and Instance-Dependent Regret Bounds

Jiayi Huang[1]     Han Zhong[1]     Liwei Wang[1,2]     Lin F. Yang[3]

[1]Center for Data Science, Peking University
[2]National Key Laboratory of General Artificial Intelligence,
School of Intelligence Science and Technology, Peking University
[3]University of California, Los Angles
{jyhuang, hanzhong}@stu.pku.edu.cn,
wanglw@cis.pku.edu.cn, linyang@ee.ucla.edu

## Abstract

While numerous works have focused on devising efficient algorithms for reinforcement learning (RL) with uniformly bounded rewards, it remains an open question whether sample or time-efficient algorithms for RL with large state-action space exist when the rewards are *heavy-tailed*, i.e., with only finite $(1 + \epsilon)$-th moments for some $\epsilon \in (0, 1]$. In this work, we address the challenge of such rewards in RL with linear function approximation. We first design an algorithm, HEAVY-OFUL, for heavy-tailed linear bandits, achieving an *instance-dependent $T$-round regret* of $\widetilde{O}\big(dT^{\frac{1-\epsilon}{2(1+\epsilon)}} \sqrt{\sum_{t=1}^{T} \nu_t^2} + dT^{\frac{1-\epsilon}{2(1+\epsilon)}}\big)$, the *first* of this kind. Here, $d$ is the feature dimension, and $\nu_t^{1+\epsilon}$ is the $(1 + \epsilon)$-th central moment of the reward at the $t$-th round. We further show the above bound is minimax optimal when applied to the worst-case instances in stochastic and deterministic linear bandits. We then extend this algorithm to the RL settings with linear function approximation. Our algorithm, termed as HEAVY-LSVI-UCB, achieves the *first* computationally efficient *instance-dependent $K$-episode regret* of $\widetilde{O}(d\sqrt{H\mathcal{U}^*}K^{\frac{1}{1+\epsilon}} + d\sqrt{H\mathcal{V}^*K})$. Here, $H$ is length of the episode, and $\mathcal{U}^*, \mathcal{V}^*$ are instance-dependent quantities scaling with the central moment of reward and value functions, respectively. We also provide a matching minimax lower bound $\Omega(dHK^{\frac{1}{1+\epsilon}} + d\sqrt{H^3K})$ to demonstrate the optimality of our algorithm in the worst case. Our result is achieved via a novel robust self-normalized concentration inequality that may be of independent interest in handling heavy-tailed noise in general online regression problems.

## 1 Introduction

Designing efficient reinforcement learning (RL) algorithms for large state-action space is a significant challenge within the RL community. A crucial aspect of RL is understanding the reward functions, which directly impacts the quality of the agent's policy. In certain real-world situations, reward distributions may exhibit heavy-tailed behavior, characterized by the occurrence of extremely large values at a higher frequency than expected in a normal distribution. Examples include image noise in signal processing [14], stock price fluctuations in financial markets [12, 18], and value functions in online advertising [11, 19]. However, much of the existing RL literature assumes rewards to be either uniformly bounded or light-tailed (e.g., sub-Gaussian). In such light-tailed settings, the primary challenge lies in learning the transition probabilities, leading most studies to assume deterministic rewards for ease of analysis [5, 22, 15]. As we will demonstrate, the complexity of learning reward

functions may dominate in heavy-tailed settings. Consequently, the performance of traditional algorithms may decline, emphasizing the need for the development of new, efficient algorithms specifically designed to handle heavy-tailed rewards.

Heavy-tailed distributions have been extensively studied in the field of statistics [9, 27] and in more specific online learning scenarios, such as bandits [7, 28, 31, 34, 40]. However, there is a dearth of theoretical research in RL concerning heavy-tailed rewards, whose distributions only admit finite $(1 + \epsilon)$-th moment for some $\epsilon \in (0, 1]$. One notable exception is Zhuang and Sui [44], which made a pioneering effort in establishing worst-case regret guarantees in *tabular* Markov Decision Processes (MDPs) with heavy-tailed rewards. However, their algorithm cannot handle RL settings with large state-action space. Moreover, their reliance on truncation-based methods is sub-optimal as these methods heavily depend on raw moments, which do not vanish in deterministic cases. Therefore, a natural question arises:

> *Can we derive sample and time-efficient algorithms for RL with large state-action space that achieve instance-dependent regret in the presence of heavy-tailed rewards?*

In this work, we focus on linear MDPs [35, 22] with heavy-tailed rewards and answer the above question affirmatively. We say a distribution is *heavy-tailed* if it only admits finite $(1 + \epsilon)$-th moment for some $\epsilon \in (0, 1]$. Our contributions are summarized as follows.

- We first propose a computationally efficient algorithm HEAVY-OFUL for heavy-tailed linear bandits. Such a setting can be regarded as a special case of linear MDPs. HEAVY-OFUL achieves an *instance-dependent* $T$-round regret of $\widetilde{O}\big(dT^{\frac{1-\epsilon}{2(1+\epsilon)}} \sqrt{\sum_{t=1}^{T} \nu_t^2} + dT^{\frac{1-\epsilon}{2(1+\epsilon)}}\big)$, the *first* of this kind. Here $d$ is the feature dimension and $\nu_t^{1+\epsilon}$ is the $(1 + \epsilon)$-th central moment of the reward at the $t$-th round. The instance-dependent regret bound has a main term that only depends on the summation of central moments, and therefore does not have a $\sqrt{T}$ term. Our regret bound is shown to be minimax optimal in both stochastic and deterministic linear bandits (See Remark 4.2 for details).

- We then extend this algorithm to time-inhomogeneous linear MDPs with heavy-tailed rewards, resulting in a new computationally efficient algorithm HEAVY-LSVI-UCB, which achieves a $K$-episode regret scaling as $\widetilde{O}(d\sqrt{H\mathcal{U}^*}K^{\frac{1}{1+\epsilon}} + d\sqrt{H\mathcal{V}^*K})$ for the *first* time. Here, $H$ is the length of the episode and $\mathcal{U}^*, \mathcal{V}^*$ are quantities measuring the central moment of the reward functions and transition probabilities, respectively (See Theorem 5.2 for details). Our regret bound is *instance-dependent* since the main term only relies on the instance-dependent quantities, which vanishes when the dynamics and rewards are deterministic. When specialized to special cases, our instance-dependent regret recovers the variance-aware regret in Li and Sun [26] (See Remark 5.3 for details) and improves existing first-order regret bounds [33, 26] (See Corollary 5.6 for details).

- We provide a minimax regret lower bound $\Omega(dHK^{\frac{1}{1+\epsilon}} + d\sqrt{H^3K})$ for linear MDPs with heavy-tailed rewards, which matches the worst-case regret bound implied by our instance-dependent regret, thereby demonstrating the minimax optimality of HEAVY-LSVI-UCB in the worst case.

For better comparisons between our algorithms and state-of-the-art results, we summarize the regrets in Table 1 and 2 for linear bandits and linear MDPs, respectively. More related works are deferred to Appendix A. Remarkably, our results demonstrate that $\epsilon = 1$ (i.e. finite variance) is sufficient to obtain variance-aware regret bounds of the same order as the case where rewards are uniformly bounded for both linear bandits and linear MDPs. The main technique contribution behind our results is a novel robust self-normalized concentration inequality inspired by Sun et al. [32]. To be more specific, it is a non-trivial generalization of adaptive Huber regression from independent and identically distributed (i.i.d.) case to heavy-tailed online regression settings and gives a *self-normalized* bound instead of the $\ell_2$-norm bound in Sun et al. [32]. Our result is computationally efficient and only scales with the feature dimension, $d$, $(1 + \epsilon)$-th *central* moment of the noise, $\nu$, and does not depend on the absolute magnitude as in other self-normalized concentration inequalities [42, 41].

**Road Map** The rest of the paper is organized as follows. Section 2 introduces heavy-tailed linear bandits and linear MDPs. Section 3 presents the robust self-normalized concentration inequality for general online regression problems with heavy-tailed noise. Section 4 and 5 give the main results for heavy-tailed linear bandits and linear MDPs, respectively. We then conclude in Section 6. Related work, experiments and all proofs can be found in Appendix.

Table 1: Comparisons with previous works on linear bandits. $d$, $T$, $\{\sigma_t\}_{t\in[T]}$, $\{\nu_t\}_{t\in[T]}$ are feature dimension, the number of rounds, the variance or central moment of the reward at the $t$-th round.

| Algorithm | Regret | Instance-dependent? | Minimax Optimal? | Deterministic-Optimal? | Heavy-Tailed Rewards? |
|---|---|---|---|---|---|
| OFUL [1] | $\widetilde{O}\left(d\sqrt{T}\right)$ | No | **Yes** | No | No |
| IDS-UCB [24] Weighted OFUL+ [41] AdaOFUL [26] | $\widetilde{O}\left(d\sqrt{\sum_{t=1}^{T}\sigma_t^2}+d\right)$ | **Yes** | **Yes** | **Yes** | No No $\epsilon=1$ |
| MENU and TOFU [31] | $\widetilde{O}\left(dT^{\frac{1}{1+\epsilon}}\right)$ | No | **Yes** | No | **Yes** |
| HEAVY-OFUL (**Ours**) | $\widetilde{O}\left(dT^{\frac{1-\epsilon}{2(1+\epsilon)}}\sqrt{\sum_{t=1}^{T}\nu_t^2}+dT^{\frac{1-\epsilon}{2(1+\epsilon)}}\right)$ | **Yes** | **Yes** | **Yes** | **Yes** |

Table 2: Comparisons with previous works on time-inhomogeneous linear MDPs. $d$, $H$, $K$, $V_1^*$, $\mathcal{G}^*$ are feature dimension, the length of the episode, the number of episodes, optimal value function, variance-dependent quantity defined in Li and Sun [26]. $\mathcal{U}^*$, $\mathcal{V}^*$ are defined in Theorem 5.2.

| Algorithm | Regret | Central Moment-Dependent? | First-Order? | Minimax Optimal? | Computa-tionally Efficient? | Heavy-Tailed Rewards? |
|---|---|---|---|---|---|---|
| LSVI-UCB[22] | $\widetilde{O}\left(\sqrt{d^3H^4K}\right)$ | No | No | No | **Yes** | No |
| FORCE [33] | $\widetilde{O}\left(\sqrt{d^3H^3V_1^*K}\right)$ | No | **Yes** | No | No | No |
| VOQL [2] LSVI-UCB++ [15] | $\widetilde{O}\left(d\sqrt{H^3K}\right)$ | No | No | **Yes** | **Yes** | No |
| VARA [26] | $\widetilde{O}\left(d\sqrt{H\mathcal{G}^*K}\right)$ | **Yes** | **Yes** | **Yes** | **Yes** | $\epsilon=1$ |
| HEAVY-LSVI-UCB (**Ours**) | $\widetilde{O}\left(d\sqrt{H\mathcal{U}^*}K^{\frac{1}{1+\epsilon}}+d\sqrt{H\mathcal{V}^*K}\right)$ | **Yes** | **Yes** | **Yes** | **Yes** | **Yes** |

**Notations** Let $\|\boldsymbol{a}\| := \|\boldsymbol{a}\|_2$. Let $[t] := \{1, 2, \ldots, t\}$. Let $\mathcal{B}_d(r) := \{\boldsymbol{x} \in \mathbb{R}^d | \|\boldsymbol{x}\| \le r\}$. Let $x_{[a,b]} := \max\{a, \min\{x, b\}\}$ denote the projection of $x$ onto the close interval $[a, b]$. Let $\sigma(\{X_s\}_{s\in[t]})$ be the $\sigma$-field generated by random vectors $\{X_s\}_{s\in[t]}$.

## 2 Preliminaries

### 2.1 Heavy-Tailed Linear Bandits

**Definition 2.1** (Heterogeneous linear bandits with heavy-tailed rewards)**.** Let $\{\mathcal{D}_t\}_{t\ge 1}$ denote a series of fixed decision sets, where all $\boldsymbol{\phi}_t \in \mathcal{D}_t$ satisfy $\|\boldsymbol{\phi}_t\| \le L$ for some known upper bound $L$. At each round $t$, the agent chooses $\boldsymbol{\phi}_t \in \mathcal{D}_t$, then receives a reward $R_t$ from the environment. We define the filtration $\{\mathcal{F}_t\}_{t\ge 1}$ as $\mathcal{F}_t = \sigma(\{\boldsymbol{\phi}_s, R_s\}_{s\in[t]} \cup \{\boldsymbol{\phi}_{t+1}\})$ for any $t \ge 1$. We assume $R_t = \langle \boldsymbol{\phi}_t, \boldsymbol{\theta}^* \rangle + \varepsilon_t$ with the unknown coefficient $\boldsymbol{\theta}^* \in \mathcal{B}_d(B)$ for some known upper bound $B$. The random variable $\varepsilon_t \in \mathbb{R}$ is $\mathcal{F}_t$-measurable and satisfies $\mathbb{E}[\varepsilon_t | \mathcal{F}_{t-1}] = 0, \mathbb{E}[|\varepsilon_t|^{1+\epsilon} | \mathcal{F}_{t-1}] = \nu_t^{1+\epsilon}$ for some $\epsilon \in (0, 1]$ with $\nu_t$ being $\mathcal{F}_{t-1}$-measurable.

The agent aims to minimize the $T$-round *pseudo-regret* defined as $\mathrm{Regret}(T) = \sum_{t=1}^{T}[\langle \boldsymbol{\phi}_t^*, \boldsymbol{\theta}^* \rangle - \langle \boldsymbol{\phi}_t, \boldsymbol{\theta}^* \rangle]$, where $\boldsymbol{\phi}_t^* = \mathrm{argmax}_{\boldsymbol{\phi}\in\mathcal{D}_t}\langle \boldsymbol{\phi}, \boldsymbol{\theta}^* \rangle$.

### 2.2 Linear MDPs with Heavy-Tailed Rewards

We use a tuple $M = M(\mathcal{S}, \mathcal{A}, H, \{R_h\}_{h\in[H]}, \{\mathbb{P}_h\}_{h\in[H]})$ to describe the *time-inhomogeneous finite-horizon MDP*, where $\mathcal{S}$ and $\mathcal{A}$ are state space and action space, respectively, $H$ is the length of the episode, $R_h : \mathcal{S} \times \mathcal{A} \to \mathbb{R}$ is the random reward function with expectation $r_h : \mathcal{S} \times \mathcal{A} \to \mathbb{R}$, and $\mathbb{P}_h : \mathcal{S} \times \mathcal{A} \to \Delta(\mathcal{S})$ is the transition probability function. More details can be found in Puterman [30]. A time-dependent *policy* $\pi = \{\pi_h\}_{h\in H}$ satisfies $\pi_h : \mathcal{S} \to \Delta(\mathcal{A})$ for any $h \in [H]$. When the policy is deterministic, we use $\pi_h(s_h)$ to denote the action chosen at the $h$-th step given $s_h$ by policy $\pi$. For any state-action pair $(s, a) \in \mathcal{S} \times \mathcal{A}$, we define the *state-action value function* $Q_h^\pi(s, a)$ and *state value function* $V_h^\pi(s)$ as follows: $Q_h^\pi(s, a) = \mathbb{E}\left[\sum_{h'=h}^{H} r(s_{h'}, a_{h'}) | s_h = s, a_h = a\right]$, $V_h^\pi(s) = Q_h^\pi(s, \pi_h(s))$, where the expectation is taken with respect to the transition probability of

$M$ and the agent's policy $\pi$. If $\pi$ is randomized, then the definition of $V$ should have an expectation. Denote the optimal value functions as $V_h^*(s) = \sup_\pi V_h^\pi(s)$ and $Q_h^*(s,a) = \sup_\pi Q_h^\pi(s,a)$.

We introduce the following shorthands for simplicity. At the $h$-th step, for any value function $V : \mathcal{S} \to \mathbb{R}$, let $[\mathbb{P}_h V](s,a) = \mathbb{E}_{s' \sim \mathbb{P}_h(\cdot|s,a)} V(s')$, $[\mathbb{V}_h V](s,a) = [\mathbb{P}_h V^2](s,a) - [\mathbb{P}_h V]^2(s,a)$ denote the expectation and the variance of the next-state value function at the $h$-th step given $(s,a)$.

We aim to minimize the $K$-episode *regret* defined as $\text{Regret}(K) = \sum_{k=1}^K [V_1^*(s_1^k) - V_1^{\pi^k}(s_1^k)]$.

In the rest of this section, we introduce linear MDPs with heavy-tailed rewards. We first give the definition of linear MDPs studied in Yang and Wang [35], Jin et al. [22], with emphasis that the rewards in their settings are deterministic or uniformly bounded. Then we focus on the heavy-tailed random rewards.

**Definition 2.2.** An MDP $M = M(\mathcal{S}, \mathcal{A}, H, \{R_h\}_{h\in[H]}, \{\mathbb{P}_h\}_{h\in[H]})$ is a *time-inhomogeneous finite-horizon linear MDP*, if there exist known feature maps $\phi(s,a) : \mathcal{S} \times \mathcal{A} \to \mathcal{B}_d(1)$, unknown $d$-dimensional signed measures $\{\boldsymbol{\mu}_h^*\}_{h\in[H]}$ over $\mathcal{S}$ with $\|\boldsymbol{\mu}_h^*(\mathcal{S})\| := \int_{s\in\mathcal{S}} |\boldsymbol{\mu}(s)| \mathrm{d}s \le \sqrt{d}$ and unknown coefficients $\{\boldsymbol{\theta}_h^*\}_{h\in[H]} \subseteq \mathcal{B}_d(B)$ for some known upper bound $B$ such that
$$r_h(s,a) = \langle \phi(s,a), \boldsymbol{\theta}_h^* \rangle, \quad \mathbb{P}_h(\cdot|s,a) = \langle \phi(s,a), \boldsymbol{\mu}_h^*(\cdot) \rangle$$
for any state-action pair $(s,a) \in \mathcal{S} \times \mathcal{A}$ and timestep $h \in [H]$.

**Assumption 2.3** (Realizable rewards). For all $(s,a,h) \in \mathcal{S} \times \mathcal{A} \times [H]$, the random reward $R_h(s,a)$ is independent of next state $s' \sim \mathbb{P}_h(\cdot|s,a)$ and admits the linear structure
$$R_h(s,a) = \langle \phi(s,a), \boldsymbol{\theta}_h^* \rangle + \varepsilon_h(s,a),$$
where $\varepsilon_h(s,a)$ is a mean-zero heavy-tailed random variable specified below.

We introduce the notation $\boldsymbol{\nu}_n[X] = \mathbb{E}[|X - \mathbb{E}X|^n]$ for the $n$-th central moment of any random variable $X$. And for any random reward function at the $h$-th step $R_h : \mathcal{S} \times \mathcal{A} \to \mathbb{R}$, let
$$[\mathbb{E}_h R_h](s,a) = \mathbb{E}[R_h(s_h,a_h)|(s_h,a_h) = (s,a)],$$
$$[\boldsymbol{\nu}_{1+\epsilon} R_h](s,a) = \mathbb{E}[|[R_h - \mathbb{E}_h R_h](s_h,a_h)|^{1+\epsilon}|(s_h,a_h) = (s,a)]$$
denote its expectation and the $(1+\epsilon)$-th central moment given $(s_h,a_h) = (s,a)$ for short.

**Assumption 2.4** (Heavy-tailedness of rewards). Random variable $\varepsilon_h(s,a)$ satisfies $[\mathbb{E}_h \varepsilon_h](s,a) = 0$. And for some known $\epsilon, \epsilon' \in (0,1]$ and constants $\nu_R, \nu_{R^\epsilon} \ge 0$, the following unknown moments of $\varepsilon_h(s,a)$ satisfy
$$[\mathbb{E}_h |\varepsilon_h|^{1+\epsilon}](s,a) \le \nu_R^{1+\epsilon}, \quad [\boldsymbol{\nu}_{1+\epsilon'} |\varepsilon_h|^{1+\epsilon}](s,a) \le \nu_{R^\epsilon}^{1+\epsilon'}$$
for all $(s,a,h) \in \mathcal{S} \times \mathcal{A} \times [H]$.

Assumption 2.4 generalizes Assumption 2.2 of Li and Sun [26], which is the weakest moment condition on random rewards in the current literature of RL with function approximation. Setting $\epsilon = 1$ and $\epsilon' = 1$ immediately recovers their settings.

**Assumption 2.5** (Realizable central moments). There are some unknown coefficients $\{\boldsymbol{\psi}_h^*\}_{h\in[H]} \subseteq \mathcal{B}_d(W)$ for some known upper bound $W$ such that
$$[\mathbb{E}_h |\varepsilon_h|^{1+\epsilon}](s,a) = \langle \phi(s,a), \boldsymbol{\psi}_h^* \rangle$$
for all $(s,a,h) \in \mathcal{S} \times \mathcal{A} \times [H]$.

**Remark 2.6.** When $\epsilon = 1$, that is the rewards have finite variance, Li and Sun [26] use the fact that $[\boldsymbol{\nu}_2 R_h](s,a) = [\mathbb{V}_h R_h](s,a) = [\mathbb{E}_h R_h^2](s,a) - [\mathbb{E}_h R_h]^2(s,a)$, assume the linear realizability of the second moment $[\mathbb{E}_h R_h^2](s,a)$, and estimate it instead. However, when $\epsilon < 1$, there is no such relationship between the $(1+\epsilon)$-th central moment $[\boldsymbol{\nu}_{1+\epsilon} R_h](s,a)$ and the $(1+\epsilon)$-th raw moment $[\mathbb{E}_h R_h^{1+\epsilon}](s,a)$. Thus, we adopt a new approach to estimate $[\boldsymbol{\nu}_{1+\epsilon} R_h](s,a)$ directly, and bound the error by a novel perturbation analysis of adaptive Huber regression in Appendix C.3.

**Assumption 2.7** (Bounded cumulative rewards). For any policy $\pi$, let $\{s_h, a_h, R_h\}_{h\in[H]}$ be a random trajectory following policy $\pi$. And define $r_\pi = \sum_{h=1}^H [\mathbb{E}_h R_h](s_h, a_h) = \sum_{h=1}^H r_h(s_h, a_h)$. We assume (1) $0 \le r_\pi \le \mathcal{H}$. (2) $\sum_{h=1}^H [\boldsymbol{\nu}_{1+\epsilon} R_h]^{\frac{2}{1+\epsilon}}(s_h, a_h) \le \mathcal{U}$. (3) $\text{Var}(r_\pi) \le \mathcal{V}$.

Here, (1) gives an upper bound of cumulative expected rewards $r_\pi$. (2) assumes the summation of $(1+\epsilon)$-th central moment of rewards $[\boldsymbol{\nu}_{1+\epsilon} R_h](s_h, a_h)$ is bounded since $[\sum_{h=1}^H [\boldsymbol{\nu}_{1+\epsilon} R_h](s_h, a_h)]^{\frac{2}{1+\epsilon}} \le \sum_{h=1}^H [\boldsymbol{\nu}_{1+\epsilon} R_h]^{\frac{2}{1+\epsilon}}(s_h, a_h) \le \mathcal{U}$ due to Jensen's inequality. And (3) is to bound the variance of $r_\pi$ along the trajectory following policy $\pi$.

---

**Algorithm 1** Adaptive Huber Regression

---

**Require:** Number of total rounds $T$, confidence level $\delta$, regularization parameter $\lambda$, $\sigma_{\min}$, parameters for adaptive Huber regression $c_0, c_1, \tau_0$, estimated central moment $\widehat{\nu}_t$ and moment parameter $b$ that satisfy $\nu_t/\widehat{\nu}_t \leq b$ for all $t \leq T$.

**Ensure:** The estimated coefficient $\boldsymbol{\theta}_t$.

1: $\kappa = d \log(1 + TL^2/(d\lambda\sigma_{\min}^2))$.

2: Set $\boldsymbol{H}_{t-1} = \lambda\boldsymbol{I} + \sum_{s=1}^{t-1} \sigma_s^{-2}\boldsymbol{\phi}_s\boldsymbol{\phi}_s^\top$.

3: Set $\sigma_t = \max\left\{\widehat{\nu}_t, \sigma_{\min}, \frac{\|\boldsymbol{\phi}_t\|_{\boldsymbol{H}_{t-1}^{-1}}}{c_0}, \frac{\sqrt{LB}}{c_1^{\frac{1}{4}}(2\kappa b^2)^{\frac{1}{4}}}\|\boldsymbol{\phi}_t\|_{\boldsymbol{H}_{t-1}^{-1}}^{\frac{1}{2}}\right\}$.

4: Set $\tau_t = \tau_0 \frac{\sqrt{1+w_t^2}}{w_t} t^{\frac{1-\epsilon}{2(1+\epsilon)}}$ with $w_t = \|\boldsymbol{\phi}_t/\sigma_t\|_{\boldsymbol{H}_{t-1}^{-1}}$.

5: Define the loss function $L_t(\boldsymbol{\theta}) := \frac{\lambda}{2}\|\boldsymbol{\theta}\|^2 + \sum_{s=1}^{t} \ell_{\tau_s}\left(\frac{y_s - \langle\boldsymbol{\phi}_s, \boldsymbol{\theta}\rangle}{\sigma_s}\right)$.

6: Compute $\boldsymbol{\theta}_t = \operatorname{argmin}_{\boldsymbol{\theta}\in\mathcal{B}_d(B)} L_t(\boldsymbol{\theta})$.

---

# 3 Adaptive Huber Regression

At the core of our algorithms for both heavy-tailed linear bandits and linear MDPs is a new approach – adaptive Huber regression – to handle heavy-tailed noise. Sun et al. [32] imposed adaptive Huber regression to handle i.i.d. heavy-tailed noise by utilizing Huber loss [17] as a surrogate of squared loss. Li and Sun [26] modified adaptive Huber regression for heterogeneous online settings, where the variances in each round are different. However, it is not readily applicable to deal with heavy-tailed noise. Our contribution in this section is to construct a new self-normalized concentration inequality for general online regression problems with heavy-tailed noise.

We first give a brief introduction to Huber loss function and its properties.

**Definition 3.1** (Huber loss). *Huber loss* is defined as

$$\ell_\tau(x) = \begin{cases} \frac{x^2}{2} & \text{if } |x| \leq \tau, \\ \tau|x| - \frac{\tau^2}{2} & \text{if } |x| > \tau, \end{cases} \tag{3.1}$$

where $\tau > 0$ is referred as a robustness parameter.

Huber loss is first proposed by Huber [17] as a robust version of squared loss while preserving the convex property. Specifically, Huber loss is a quadratic function of $x$ when $|x|$ is less than the threshold $\tau$, while becomes linearly dependent on $|x|$ when $|x|$ grows larger than $\tau$. It has the property of strongly convex near zero point and is not sensitive to outliers. See Appendix C.1 for more properties of Huber loss.

Next, we define general online regression problems with heavy-tailed noise, which include heavy-tailed linear bandits as a special case. Then we utilize Huber loss to estimate $\boldsymbol{\theta}^*$. Below we give the main theorem to bound the deviation of the estimated $\boldsymbol{\theta}_t$ in Algorithm 1 from the ground truth $\boldsymbol{\theta}^*$.

**Definition 3.2.** Let $\{\mathcal{F}_t\}_{t\geq 1}$ be a filtration. For all $t > 0$, let random variables $y_t, \varepsilon_t$ be $\mathcal{F}_t$-measurable and random vector $\boldsymbol{\phi}_t \in \mathcal{B}_d(L)$ be $\mathcal{F}_{t-1}$-measurable. Suppose $y_t = \langle\boldsymbol{\phi}_t, \boldsymbol{\theta}^*\rangle + \varepsilon_t$, where $\boldsymbol{\theta}^* \in \mathcal{B}_d(B)$ is an unknown coefficient and

$$\mathbb{E}[\varepsilon_t|\mathcal{F}_{t-1}] = 0, \quad \mathbb{E}[|\varepsilon_t|^{1+\epsilon}|\mathcal{F}_{t-1}] = \nu_t^{1+\epsilon}$$

for some $\epsilon \in (0, 1]$. The goal is to estimate $\boldsymbol{\theta}^*$ at any round $t$ given the realizations of $\{\boldsymbol{\phi}_s, y_s\}_{s\in[t]}$.

**Theorem 3.3.** For the online regression problems in Definition 3.2, we solve for $\boldsymbol{\theta}_t$ by adaptive Huber regression in Algorithm 1 with $c_0, c_1, \tau_0$ in Appendix C.2. Then for any $\delta \in (0, 1)$, with probability at least $1 - 3\delta$, for all $t \leq T$, we have $\|\boldsymbol{\theta}_t - \boldsymbol{\theta}^*\|_{\boldsymbol{H}_t} \leq \beta_t$, where $\boldsymbol{H}_t$ is defined in Algorithm 1 and

$$\beta_t = 3\sqrt{\lambda}B + 24t^{\frac{1-\epsilon}{2(1+\epsilon)}}\sqrt{2\kappa}b(\log 3T)^{\frac{1-\epsilon}{2(1+\epsilon)}}(\log(2T^2/\delta))^{\frac{\epsilon}{1+\epsilon}}. \tag{3.2}$$

*Proof.* To derive a tight high-probability bound, we take the most advantage of the properties of Huber loss. A Chernoff bounding technique is used to bound the main error term, which requires a careful analysis of the moment generating function. See Appendix C.2 for a detailed proof. □

---

**Algorithm 2** HEAVY-OFUL

---

**Require:** Number of total rounds $T$, confidence level $\delta$, regularization parameter $\lambda$, $\sigma_{\min}$, parameters for adptive Huber regression $c_0, c_1, \tau_0$, confidence radius $\beta_t$.

1: $\kappa = d \log(1 + \frac{TL^2}{d\lambda\sigma_{\min}^2}), \mathcal{C}_0 = \mathcal{B}_d(B), \boldsymbol{H}_0 = \lambda\boldsymbol{I}$.
2: **for** $t = 1, \ldots, T$ **do**
3:     Observe $\mathcal{D}_t$.
4:     Set $(\boldsymbol{\phi}_t, \cdot) = \text{argmax}_{\boldsymbol{\phi}\in\mathcal{D}_t, \boldsymbol{\theta}\in\mathcal{C}_{t-1}}\langle\boldsymbol{\phi}, \boldsymbol{\theta}\rangle$.
5:     Play $\boldsymbol{\phi}_t$ and observe $R_t, \nu_t$.
6:     Set $\sigma_t = \max\left\{\nu_t, \sigma_{\min}, \frac{\|\boldsymbol{\phi}_t\|_{\boldsymbol{H}_{t-1}^{-1}}}{c_0}, \frac{\sqrt{LB}}{c_1^{\frac{1}{4}}(2\kappa)^{\frac{1}{4}}}\|\boldsymbol{\phi}_t\|_{\boldsymbol{H}_{t-1}^{-1}}^{\frac{1}{2}}\right\}$.
7:     Set $\tau_t = \tau_0 \frac{\sqrt{1+w_t^2}}{w_t}t^{\frac{1-\epsilon}{2(1+\epsilon)}}$ with $w_t = \|\boldsymbol{\phi}_t/\sigma_t\|_{\boldsymbol{H}_{t-1}^{-1}}$.
8:     Update $\boldsymbol{H}_t = \boldsymbol{H}_{t-1} + \sigma_t^{-2}\boldsymbol{\phi}_t\boldsymbol{\phi}_t^\top$.
9:     Solve for $\boldsymbol{\theta}_t$ by Algorithm 1 and set $\mathcal{C}_t = \{\boldsymbol{\theta}\in\mathbb{R}^d | \|\boldsymbol{\theta} - \boldsymbol{\theta}_t\|_{\boldsymbol{H}_t} \leq \beta_t\}$.
10: **end for**

---

We refer to the regression process in Line 6 of Algorithm 1 as *adaptive Huber regression* in line with Sun et al. [32] to emphasize that the value of robustness parameter $\tau_t$ is chosen to adapt to data for a better trade-off between bias and robustness. Specifically, since we are in the online setting, $\boldsymbol{\phi}_t$ are dependent on $\{\boldsymbol{\phi}_s\}_{s<t}$, which is the key difference from the i.i.d. case in Sun et al. [32] where they set $\tau_t = \tau_0$, for all $t \leq T$. Thus, as shown in Line 4 of Algorithm 1, inspired by Li and Sun [26], we adjust $\tau_t$ according to the importance of observations $w_t = \|\boldsymbol{\phi}_t/\sigma_t\|_{\boldsymbol{H}_{t-1}^{-1}}$, where $\sigma_t$ is specified below. In the case where $\epsilon < 1$, different from Li and Sun [26], we first choose $\tau_t$ to be small for robust purposes, then gradually increase it with $t$ to reduce the bias.

Next, we illustrate the reason for setting $\sigma_t$ via Line 3 of Algorithm 1. We use $\widehat{\nu}_t \in \mathcal{F}_{t-1}$ to estimate the central moment $\nu_t$ and use moment parameter $b$ to measure the closeness between $\widehat{\nu}_t$ and $\nu_t$. When we choose $\widehat{\nu}_t$ as an upper bound of $\nu_t$, $b$ becomes a constant that equals to 1. And $\sigma_{\min}$ is a small positive constant to avoid singularity. The last two terms with respect to $c_0$ and $c_1$ are set according to the uncertainty $\|\boldsymbol{\phi}_t\|_{\boldsymbol{H}_{t-1}^{-1}}$. In addition, setting the parameter $c_0 \leq 1$ yields $w_t \leq 1$, which is essential to meet the condition of elliptical potential lemma [1].

**Remark 3.4.** The error bound $\beta_t$ in (3.2) is only related to the feature dimension $d$ and moment parameter $b$. While the Bernstein-style self-normalized concentration bounds [42, 41] depend on the magnitude of $\varepsilon_t$, thus cannot handle heavy-tailed errors.

## 4 Linear Bandits

In this section, we show the algorithm HEAVY-OFUL in Algorithm 2 for heavy-tailed linear bandits in Definition 2.1. We first give a brief algorithm description, and then provide a theoretical regret analysis.

### 4.1 Algorithm Description

HEAVY-OFUL follows the principle of Optimism in the Face of Uncertainty (OFU) [1], and uses adaptive Huber regression in Section 3 to maintain a set $\mathcal{C}_t$ that contains the unknown coefficient $\boldsymbol{\theta}^*$ with high probability. Specifically, at the $t$-th round, HEAVY-OFUL estimates the expected reward of any arm $\boldsymbol{\phi}$ as $\max_{\boldsymbol{\theta}\in\mathcal{C}_{t-1}}\langle\boldsymbol{\phi}, \boldsymbol{\theta}\rangle$, and selects the arm that maximizes the estimated reward. The agent then receives the reward $R_t$ and updates the confidence set $\mathcal{C}_t$ based on the information up to round $t$ with its center $\boldsymbol{\theta}_t$ computed by adaptive Huber regression as in Line 9 of Algorithm 2.

### 4.2 Regret Analysis

We next give the instance-dependent regret upper bound of HEAVY-OFUL in Theorem 4.1.

**Theorem 4.1.** For the heavy-tailed linear bandits in Definition 2.1, we set $c_0, c_1, \tau_0, \beta_t$ in Algorithm 2 according to Theorem 3.3 with $b = 1$. Besides, let $\lambda = d/B^2$, and $\sigma_{\min} = 1/\sqrt{T}$. Then with

probability at least $1 - 3\delta$, the regret of HEAVY-OFUL is bounded by

$$\text{Regret}(T) = \widetilde{O}\left( dT^{\frac{1-\epsilon}{2(1+\epsilon)}} \sqrt{\sum_{t=1}^{T} \nu_t^2} + dT^{\frac{1-\epsilon}{2(1+\epsilon)}} \right).$$

*Proof.* The proof uses the self-normalized concentration inequality of adaptive Huber regression and a careful analysis to bound the summation of bonuses. See Appendix D.1 for a detailed proof. $\quad\square$

**Remark 4.2.** Theorem 4.1 shows HEAVY-OFUL achieves an instance-dependent regret bound. When we assume $\nu_t, \forall t \geq 1$ have uniform upper bound $\nu$ (which can be treated as a constant), then the bound is reduced to $\widetilde{O}(dT^{\frac{1}{1+\epsilon}})$. It matches the lower bound $\Omega(dT^{\frac{1}{1+\epsilon}})$ by Shao et al. [31] up to logarithmic factors. In the deterministic scenario, where $\epsilon = 1$ and $\nu_t = 0$, for all $t \geq 1$, the bound is reduced to $\widetilde{O}(d)$. It matches the lower bound $\Omega(d)$[1] up to logarithmic factors.

# 5 Linear MDPs

In this section, we show the algorithm HEAVY-LSVI-UCB in Algorithm 3 for linear MDP with heavy-tailed rewards defined in Section 2.2. Let $\phi_{k,h} := \phi(s_{k,h}, a_{k,h})$ for short. We first give the algorithm description intuitively, then provide the computational complexity and regret bound.

## 5.1 Algorithm Description

HEAVY-LSVI-UCB features a novel combination of adaptive Huber regression in Section 3 and existing algorithmic frameworks for linear MDPs with bounded rewards [22, 15]. At a high level, HEAVY-LSVI-UCB employs separate estimation techniques to handle heavy-tailed rewards and transition kernels. Specifically, we utilize adaptive Huber regression proposed in Section 3 to estimate heavy-tailed rewards and weighted ridge regression [42, 15] to estimate the expected next-state value functions. Then, it follows the value iteration scheme to update the optimistic and pessimistic estimation of the optimal value function $Q_h^k, V_h^k$ and $\check{Q}_h^k, \check{V}_h^k$, respectively, via a rare-switching policy as in Line 7 to 15 of Algorithm 3. We highlight the key steps of HEAVY-LSVI-UCB as follows.

**Estimation for expected heavy-tailed rewards** Since the expected rewards have linear structure in linear MDPs, i.e., $r_h(s,a) = \langle \phi(s,a), \theta_h^* \rangle$, we use adaptive Huber regression to estimate $\theta_h^*$:

$$\theta_{k,h} = \operatorname*{argmin}_{\theta \in \mathcal{B}_d(B)} \frac{\lambda_R}{2} \|\theta\|^2 + \sum_{i=1}^{k} \ell_{\tau_{i,h}}\left( \frac{R_{i,h} - \langle \phi_{i,h}, \theta \rangle}{\nu_{i,h}} \right), \tag{5.1}$$

where $\nu_{i,h}$ will be specified later.

**Estimation for central moment of rewards** By Assumption 2.5, the $(1+\epsilon)$-th central moment of rewards is linear in $\phi$, i.e., $[\nu_{1+\epsilon} R_h](s,a) = \langle \phi(s,a), \psi_h^* \rangle$. Motivated by this, we estimate $\psi_h^*$ by adaptive Huber regression as

$$\psi_{k,h} = \operatorname*{argmin}_{\psi \in \mathcal{B}_d(W)} \frac{\lambda_R}{2} \|\psi\|^2 + \sum_{i=1}^{k} \ell_{\widetilde{\tau}_{i,h}}\left( \frac{|\varepsilon_{i,h}|^{1+\epsilon} - \langle \phi_{i,h}, \psi \rangle}{\nu_{i,h}} \right), \tag{5.2}$$

where $W$ is the upper bound of $\|\psi_h^*\|$ defined in Assumption 2.5. Since $\varepsilon_{i,h}$ is intractable, we estimate it by $\widehat{\varepsilon}_{i,h} = R_{i,h} - \langle \phi_{i,h}, \theta_{i,h} \rangle$, which gives $\widehat{\psi}_{k,h}$ as

$$\widehat{\psi}_{k,h} = \operatorname*{argmin}_{\psi \in \mathcal{B}_d(W)} \frac{\lambda_R}{2} \|\psi\|^2 + \sum_{i=1}^{k} \ell_{\widetilde{\tau}_{i,h}}\left( \frac{|\widehat{\varepsilon}_{i,h}|^{1+\epsilon} - \langle \phi_{i,h}, \psi \rangle}{\nu_{i,h}} \right). \tag{5.3}$$

The inevitable error between $\psi_{k,h}$ and $\widehat{\psi}_{k,h}$ can be quantified by a novel perturbation analysis of adaptive Huber regression in Appendix C.3.

---

[1]Consider the decision set consisting of unit bases of $\mathbb{R}^d$. Given that each arm pull can only yield information about a single coordinate, it is inevitable that $d$ pulls are required for exploration.

**Algorithm 3** HEAVY-LSVI-UCB

**Require:** Number of episodes $K$, confidence level $\delta$, regularization parameter $\lambda_R, \lambda_V, \nu_{\min}, \sigma_{\min}$, confidence radius $\beta_{R^\epsilon}, \beta_0, \beta_R, \beta_V$.

1: $\kappa = d\log(1 + \frac{K}{d\lambda_R \nu_{\min}^2})$.
2: $\boldsymbol{\theta}_{0,h} = \widehat{\boldsymbol{w}}_{0,h} = \check{\boldsymbol{w}}_{0,h} = \mathbf{0}$, $\boldsymbol{H}_{0,h} = \lambda_R \boldsymbol{I}$, $\boldsymbol{\Sigma}_{0,h} = \lambda_V \boldsymbol{I}$, UPDATE = TRUE.
3: **for** $k = 1, \ldots, K$ **do**
4: $\quad V_{H+1}^k(\cdot) = \check{V}_{H+1}^k(\cdot) = 0$.
5: $\quad$ **for** $h = H, \ldots, 1$ **do**
6: $\qquad$ Compute $\boldsymbol{\theta}_{k-1,h}$, $\widehat{\boldsymbol{w}}_{k-1,h}$ and $\check{\boldsymbol{w}}_{k-1,h}$ via (5.1) and (5.8).
7: $\qquad$ **if** UPDATE **then**
8: $\qquad\quad Q_h^k(\cdot,\cdot) = \langle \boldsymbol{\phi}(\cdot,\cdot), \boldsymbol{\theta}_{k-1,h} + \widehat{\boldsymbol{w}}_{k-1,h}\rangle + \beta_{R,k-1}\|\boldsymbol{\phi}(\cdot,\cdot)\|_{\boldsymbol{H}_{k-1,h}^{-1}} + \beta_V\|\boldsymbol{\phi}(\cdot,\cdot)\|_{\boldsymbol{\Sigma}_{k-1,h}^{-1}}$.
9: $\qquad\quad \check{Q}_h^k(\cdot,\cdot) = \langle \boldsymbol{\phi}(\cdot,\cdot), \boldsymbol{\theta}_{k-1,h} + \check{\boldsymbol{w}}_{k-1,h}\rangle - \beta_{R,k-1}\|\boldsymbol{\phi}(\cdot,\cdot)\|_{\boldsymbol{H}_{k-1,h}^{-1}} - \beta_V\|\boldsymbol{\phi}(\cdot,\cdot)\|_{\boldsymbol{\Sigma}_{k-1,h}^{-1}}$.
10: $\qquad\quad Q_h^k(\cdot,\cdot) = \min\{Q_h^k(\cdot,\cdot), Q_h^{k-1}(\cdot,\cdot), \mathcal{H}\}$, $\check{Q}_h^k(\cdot,\cdot) = \max\{\check{Q}_h^k(\cdot,\cdot), \check{Q}_h^{k-1}(\cdot,\cdot), 0\}$.
11: $\qquad\quad$ Set $k_{\text{last}} = k$.
12: $\qquad$ **else**
13: $\qquad\quad Q_h^k(\cdot,\cdot) = Q_h^{k-1}(\cdot,\cdot)$, $\check{Q}_h^k(\cdot,\cdot) = \check{Q}_h^{k-1}(\cdot,\cdot)$.
14: $\qquad$ **end if**
15: $\qquad V_h^k(\cdot) = \max_a Q_h^k(\cdot,a)$, $\check{V}_h^k(\cdot) = \max_a \check{Q}_h^k(\cdot,a)$, $\pi_h^k(\cdot) = \arg\max_a Q_h^k(\cdot,a)$.
16: $\quad$ **end for**
17: $\quad$ Observe initial state $s_{k,1}$.
18: $\quad$ **for** $h = 1, \ldots, H$ **do**
19: $\qquad$ Take action $a_{k,h} = \pi_h^k(s_{k,h})$ and observe $R_{k,h}, s_{k,h+1}$.
20: $\qquad$ Set $\nu_{k,h}$ and $\sigma_{k,h}$ according to (5.4) and (5.9) respectively.
21: $\qquad$ Set $\tau_{k,h} = \tau_0 \frac{\sqrt{1+w_{k,h}^2}}{w_{k,h}} k^{\frac{1-\epsilon}{2(1+\epsilon)}}$, $\widetilde{\tau}_{k,h} = \widetilde{\tau}_0 \frac{\sqrt{1+w_{k,h}^2}}{w_{k,h}} k^{\frac{1-\epsilon'}{2(1+\epsilon')}}$ with $w_{k,h} = \|\boldsymbol{\phi}_{k,h}/\nu_{k,h}\|_{\boldsymbol{H}_{k,h}^{-1}}$.
22: $\qquad$ Update $\boldsymbol{H}_{k,h} = \boldsymbol{H}_{k-1,h} + \frac{1}{\nu_{k,h}^2}\boldsymbol{\phi}_{k,h}\boldsymbol{\phi}_{k,h}^\top$ and $\boldsymbol{\Sigma}_{k,h} = \boldsymbol{\Sigma}_{k-1,h} + \frac{1}{\sigma_{k,h}^2}\boldsymbol{\phi}_{k,h}\boldsymbol{\phi}_{k,h}^\top$.
23: $\quad$ **end for**
24: $\quad$ **if** $\exists h' \in [H]$ such that $\det(\boldsymbol{H}_{k,h'}) \geq 2\det(\boldsymbol{H}_{k_{\text{last}},h'})$ or $\det(\boldsymbol{\Sigma}_{k,h'}) \geq 2\det(\boldsymbol{\Sigma}_{k_{\text{last}},h'})$ **then**
25: $\qquad$ Set UPDATE = TRUE.
26: $\quad$ **else**
27: $\qquad$ Set UPDATE = FALSE.
28: $\quad$ **end if**
29: **end for**

We then set the weight $\nu_{k,h}$ for adaptive Huber regression as

$$\nu_{k,h} = \max\left\{\widehat{\nu}_{k,h}, \nu_{\min}, \frac{\|\boldsymbol{\phi}_{k,h}\|_{\boldsymbol{H}_{k-1,h}^{-1}}}{c_0}, \frac{\sqrt{\max\{B,W\}}}{c_1^{\frac{1}{4}}(2\kappa)^{\frac{1}{4}}}\|\boldsymbol{\phi}_{k,h}\|_{\boldsymbol{H}_{k-1,h}^{-1}}^{\frac{1}{2}}\right\}, \tag{5.4}$$

where $\nu_{\min}$ is a small positive constant to avoid the singularity, $\widehat{\nu}_{k,h}^{1+\epsilon} = [\widehat{\boldsymbol{\nu}}_{1+\epsilon}R_h](s_{k,h}, a_{k,h}) + W_{k,h}$ is a high-probability upper bound of rewards' central moment $[\boldsymbol{\nu}_{1+\epsilon}R_h](s_{k,h}, a_{k,h})$ with

$$[\widehat{\boldsymbol{\nu}}_{1+\epsilon}R_h](s_{k,h}, a_{k,h}) = \langle \boldsymbol{\phi}_{k,h}, \widehat{\boldsymbol{\psi}}_{k-1,h}\rangle, \tag{5.5}$$
$$W_{k,h} = (\beta_{R^\epsilon,k-1} + 6\mathcal{H}^\epsilon \beta_{R,k-1}\kappa)\|\boldsymbol{\phi}_{k,h}\|_{\boldsymbol{H}_{k-1,h}^{-1}}, \tag{5.6}$$

where $\kappa$ is defined in Algorithm 3, $\beta_{R^\epsilon,k} = \widetilde{O}(\sqrt{d}\nu_{R^\epsilon}k^{\frac{1-\epsilon'}{2(1+\epsilon')}}/\nu_{\min})$ and $\beta_{R,k} = \widetilde{O}(\sqrt{d}k^{\frac{1-\epsilon}{2(1+\epsilon)}})$.

**Estimation for expected next-state value functions** For any value function $f : \mathcal{S} \to \mathbb{R}$, we define the following notations for simplicity:

$$\boldsymbol{w}_h[f] = \int_{s\in\mathcal{S}} \boldsymbol{\mu}_h^*(s)f(s)\mathrm{d}s, \quad \widehat{\boldsymbol{w}}_{k,h}[f] = \boldsymbol{\Sigma}_{k,h}^{-1}\sum_{i=1}^k \sigma_{i,h}^{-2}\boldsymbol{\phi}_{i,h}f(s_{i,h+1}), \tag{5.7}$$

where $\sigma_{i,h}$ will be specified later. Note for any state-action pair $(s,a) \in \mathcal{S} \times \mathcal{A}$, by linear structure of transition probabilities, we have

$$[\mathbb{P}_h f](s,a) = \int_{s'\in\mathcal{S}} \langle \boldsymbol{\phi}(s,a), \boldsymbol{\mu}_h^*(s')\rangle f(s')\mathrm{d}s' = \langle \boldsymbol{\phi}(s,a), \boldsymbol{w}_h[f]\rangle.$$

In addition, for any $f, g : \mathcal{S} \to \mathbb{R}$, it holds that $\boldsymbol{w}_h[f + g] = \boldsymbol{w}_h[f] + \boldsymbol{w}_h[g]$ and $\widehat{\boldsymbol{w}}_{k,h}[f + g] = \widehat{\boldsymbol{w}}_{k,h}[f] + \widehat{\boldsymbol{w}}_{k,h}[g]$ due to the linear property of integration and ridge regression.

We remark $\widehat{\boldsymbol{w}}_{k,h}[f]$ is the estimation of $\boldsymbol{w}_h[f]$ by weighted ridge regression on $\{\boldsymbol{\phi}_{i,h}, f(s_{i,h+1})\}_{i \in [k]}$. And we estimate the coefficients $\widehat{\boldsymbol{w}}_{k,h}, \widecheck{\boldsymbol{w}}_{k,h}, \widetilde{\boldsymbol{w}}_{k,h}$

$$\widehat{\boldsymbol{w}}_{k,h} = \widehat{\boldsymbol{w}}_{k,h}[V_{h+1}^k], \quad \widecheck{\boldsymbol{w}}_{k,h} = \widehat{\boldsymbol{w}}_{k,h}[\widecheck{V}_{h+1}^k], \quad \widetilde{\boldsymbol{w}}_{k,h} = \widehat{\boldsymbol{w}}_{k,h}[(V_{h+1}^k)^2], \tag{5.8}$$

where $V_h^k$ and $\widecheck{V}_h^k$ are optimistic and pessimistic estimation of the optimal value functions.

**Estimation for variance of next-state value functions**  Inspired by He et al. [15], we set the weight $\sigma_{k,h}$ for weighted ridge regression in (5.7) as

$$\sigma_{k,h} = \max\left\{\widehat{\sigma}_{k,h}, \sqrt{d^3 H D_{k,h}}, \sigma_{\min}, \|\boldsymbol{\phi}_{k,h}\|_{\boldsymbol{\Sigma}_{k-1,h}^{-1}}, \sqrt{d^{\frac{5}{2}} H \mathcal{H}} \|\boldsymbol{\phi}_{k,h}\|_{\boldsymbol{\Sigma}_{k-1,h}^{-1}}^{\frac{1}{2}}\right\}, \tag{5.9}$$

where $\sigma_{\min}$ is a small constant to avoid singularity, $\widehat{\sigma}_{k,h}^2 = [\widehat{\mathbb{V}}_h V_{h+1}^k](s_{k,h}, a_{k,h}) + E_{k,h}$ with

$$\left[\widehat{\mathbb{V}}_h V_{h+1}^k\right](s_{k,h}, a_{k,h}) = \langle\boldsymbol{\phi}_{k,h}, \widetilde{\boldsymbol{w}}_{k-1,h}\rangle_{[0,\mathcal{H}^2]} - \langle\boldsymbol{\phi}_{k,h}, \widehat{\boldsymbol{w}}_{k-1,h}\rangle_{[0,\mathcal{H}]}^2, \tag{5.10}$$

$$E_{k,h} = \min\left\{4\mathcal{H}\langle\boldsymbol{\phi}_{k,h}, \widehat{\boldsymbol{w}}_{k-1,h} - \widecheck{\boldsymbol{w}}_{k-1,h}\rangle + 11\mathcal{H}\beta_0 \|\boldsymbol{\phi}_{k,h}\|_{\boldsymbol{\Sigma}_{k-1,h}^{-1}}, \mathcal{H}^2\right\}. \tag{5.11}$$

$$D_{k,h} = \min\left\{2\mathcal{H}\langle\boldsymbol{\phi}_{k,h}, \widehat{\boldsymbol{w}}_{k-1,h} - \widecheck{\boldsymbol{w}}_{k-1,h}\rangle + 4\mathcal{H}\beta_0 \|\boldsymbol{\phi}_{k,h}\|_{\boldsymbol{\Sigma}_{k-1,h}^{-1}}, \mathcal{H}^2\right\}, \tag{5.12}$$

where $\beta_0 = \widetilde{O}(\sqrt{d^3 H \mathcal{H}^2}/\sigma_{\min})$. Here $\widehat{\sigma}_{k,h}^2$ and $D_{k,h}$ are upper bounds of $[\mathbb{V}_h V_{h+1}^*](s_{k,h}, a_{k,h})$ and $\max\{[\mathbb{V}_h(V_{h+1}^k - V_{h+1}^*)](s_{k,h}, a_{k,h}), [\mathbb{V}_h(V_{h+1}^* - \widecheck{V}_{h+1}^k)](s_{k,h}, a_{k,h})\}$, respectively.

## 5.2 Computational Complexity

**Theorem 5.1.** For the linear MDPs with heavy-tailed rewards defined in Section 2.2, the computational complexity of HEAVY-LSVI-UCB is $\widetilde{O}(d^4|\mathcal{A}|H^3 K + HK\mathcal{R})$. Here $\mathcal{R}$ is the cost of the optimization algorithm for solving adaptive Huber regression in (5.1). Furthermore, we can specialize $\mathcal{R}$ by adopting the Nesterov accelerated method, which gives $\mathcal{R} = \widetilde{O}(d + d^{-\frac{1-\epsilon}{2(1+\epsilon)}} H^{\frac{1-\epsilon}{2(1+\epsilon)}} K^{\frac{1+2\epsilon}{2(1+\epsilon)}})$.

*Proof.* See Appendix E for a detailed proof. □

Such a complexity allows us to focus on the complexity introduced by the RL algorithm rather than the optimization subroutine for solving adaptive Huber regression. Compared to that of LSVI-UCB++ [15], $\widetilde{O}(d^4|\mathcal{A}|H^3 K)$, the extra term $\widetilde{O}(HK\mathcal{R})$ causes a slightly worse computational time in terms of $K$. This is due to the absence of a closed-form solution of adaptive Huber regression in (5.1). Thus extra optimization steps are unavoidable. Nevertheless, Nesterov accelerated method gives $\mathcal{R} = \widetilde{O}\left(K^{\frac{1+2\epsilon}{2(1+\epsilon)}}\right)$ with respect to $K$, which implies the computational complexity of HEAVY-LSVI-UCB is better than that of LSVI-UCB [22], $\widetilde{O}(d^2|\mathcal{A}|HK^2)$ in terms of $K$, thanks to the rare-switching updating policy. We conduct numerical experiments in Appendix B to further corroborate the computational efficiency of adaptive Huber regression.

## 5.3 Regret Bound

**Theorem 5.2** (Informal)**.** For the linear MDPs with heavy-tailed rewards defined in Section 2.2, we set parameters in Algorithm 3 as follows: $\lambda_R = d/\max\{B^2, W^2\}, \lambda_V = 1/\mathcal{H}^2, \nu_{\min}, \sigma_{\min}, c_0, c_1, \tau_0, \widetilde{\tau}_0, \beta_{R^\epsilon}, \beta_0, \beta_R, \beta_V$ in Appendix F.1. Then for any $\delta \in (0, 1)$, with probability at least $1 - 16\delta$, the regret of HEAVY-LSVI-UCB is bounded by

$$\text{Regret}(K) = \widetilde{O}\left(d\sqrt{H\mathcal{U}^*}K^{\frac{1}{1+\epsilon}} + d\sqrt{H\mathcal{V}^*K}\right),$$

where $\epsilon \in (0, 1], \mathcal{U}^* = \min\{\mathcal{U}_0^*, \mathcal{U}\}, \mathcal{V}^* = \min\{\mathcal{V}_0^*, \mathcal{V}\}$ with $\mathcal{U}_0^*, \mathcal{V}_0^*$ defined in Appendix F.2 and $\mathcal{H}, \mathcal{U}, \mathcal{V}$ defined in Assumption 2.7.

*Proof.* See Appendix F.2 for a formal version of Theorem 5.2 and its detailed proof. □

**Quantities $\mathcal{U}^*$, $\mathcal{V}^*$**   We make a few explanations for the quantities $\mathcal{U}^*$, $\mathcal{V}^*$. On one hand, $\mathcal{U}^*$ is upper bounded by $\mathcal{U}$, which is the upper bound of the sum of the $(1 + \epsilon)$-th central moments of reward functions along a single trajectory. On the other hand, $\mathcal{U}^*$ is no more than $\mathcal{U}_0^*$, which is the sum of the $(1 + \epsilon)$-th central moments with respect to the averaged occupancy measure of the first $K$ episodes. $\mathcal{V}^*$ is defined similar to $\mathcal{U}^*$, but measures the randomness of transition probabilities.

**Remark 5.3.** When $\epsilon = 1$, we can show that this regret is bounded by $\widetilde{O}(d\sqrt{H\mathcal{G}^*K})$, where $\mathcal{G}^*$ is an variance-dependent quantity defined by Li and Sun [26]. Thus, our result recovers their variance-aware regret bound. See Remark F.15 in Appendix F.2 for a detailed proof.

To demonstrate the optimality of our results and establish connections with existing literature, we can specialize Theorem 5.2 to obtain the worst-case regret [22, 2, 15] and first-order regret [33].

**Corollary 5.4** (Worst-case regret)**.** For the linear MDPs with heavy-tailed rewards defined in Section 2.2 and for any $\delta \in (0, 1)$, with probability at least $1 - 16\delta$, the regret of HEAVY-LSVI-UCB is bounded by

$$\widetilde{O}(dHK^{\frac{1}{1+\epsilon}} + d\sqrt{H^3K}).$$

*Proof.* Notice $\mathcal{U}^*$ and $\mathcal{V}^*$ are upper bounded by $H\nu_R^2$ and $\mathcal{H}^2$ (total variance lemma in Jin et al. [21]) respectively. When $\mathcal{H} = H$, and we treat $\nu_R$ as a constant, the result follows.   □

Next, we give the regret lower bound of linear MDPs with heavy-tailed rewards in Theorem 5.5, which shows our proposed HEAVY-LSVI-UCB is minimax optimal in the worst case.

**Theorem 5.5.** For any algorithm, there exists an $H$-episodic, $d$-dimensional linear MDP with heavy-tailed rewards such that for any $K$, the algorithm's regret is

$$\Omega(dHK^{\frac{1}{1+\epsilon}} + d\sqrt{H^3K}).$$

*Proof.* Intuitively, the proof of Theorem 5.5 follows from a combination of the lower bound constructions for heavy-tailed linear bandits in Shao et al. [31] and linear MDPs in Zhou et al. [42]. See Appendix G for a detailed proof.   □

Theorem 5.5 shows that for sufficiently large $K$, the reward term dominates in the regret bound. Thus, in heavy-tailed settings, the main difficulty is learning the reward functions.

**Corollary 5.6** (First-order regret)**.** For the linear MDPs with heavy-tailed rewards defined in Section 2.2 and for any $\delta \in (0, 1)$, with probability at least $1 - 16\delta$, the regret of HEAVY-LSVI-UCB is bounded by

$$\widetilde{O}(d\sqrt{H\mathcal{U}^*}K^{\frac{1}{1+\epsilon}} + d\sqrt{H\mathcal{H}V_1^*K}).$$

And when the rewards are uniformly bounded in $[0, 1]$, the result is reduced to the first-order regret bound of $\widetilde{O}(d\sqrt{H^2V_1^*K})$.

*Proof.* See Section F.3 for a detailed proof.   □

Our first-order regret $\widetilde{O}(d\sqrt{H^2V_1^*K})$ is minimax optimal in the worst case since $V_1^* \leq H$. And it improves the state-of-the-art result $\widetilde{O}(d\sqrt{H^3V_1^*K})$ [26] by a factor of $\sqrt{H}$.

# 6   Conclusion

In this work, we propose two computationally efficient algorithms for heavy-tailed linear bandits and linear MDPs, respectively. Our proposed algorithms, termed as HEAVY-OFUL and HEAVY-LSVI-UCB, are based on a novel self-normalized concentration inequality for adaptive Huber regression, which may be of independent interest. HEAVY-OFUL and HEAVY-LSVI-UCB achieve minimax optimal and instance-dependent regret bounds scaling with the central moments. We also provide a lower bound for linear MDPs with heavy-tailed rewards to demonstrate the optimality of HEAVY-LSVI-UCB. To the best of our knowledge, we are the first to study heavy-tailed rewards in RL with function approximation and provide a new algorithm for this setting which is both statistically and computationally efficient.

## Acknowledgments

Liwei Wang is supported in part by NSF IIS 2110170, NSF DMS 2134106, NSF CCF 2212261, NSF IIS 2143493, NSF CCF 2019844, NSF IIS 2229881. Lin F. Yang is supported in part by NSF grant 2221871, and an Amazon Research Grant.

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
