# OpenReview forum: "Tackling Heavy-Tailed Rewards in Reinforcement Learning with Function Approximation: Minimax Optimal and Instance-Dependent Regret Bounds"
_NeurIPS.cc/2023/Conference — NeurIPS 2023 poster_

### Official Review · Reviewer_gEhD · 2023-07-04

**Soundness:** 2 fair
**Presentation:** 2 fair
**Contribution:** 2 fair
**Rating:** 6
**Confidence:** 4

**Summary:**

main contributions include two-folds. For heavy-tailed payoffs, design heavy-tailed linear bandits, derive the variance-dependent T-round regret; In terms of Linear MDPs, instance-dependent K-episode regret is acquired. All paper results substantially depend on Huber loss regression techniques.

**Strengths:**

For heavy-tailed linear bandits, the author proposed how to tune the robustification parameter that balances bias and robustness on the fly.
for linear MDPs with bounded rewards problem, the author employs separate estimation techniques to handle heavy-tailed rewards and transition kernels, i.e, utilizing adaptive Huber regression to estimate heavy-tailed rewards and weighted ridge regression to estimate the expected next-state value functions.

**Weaknesses:**

The derived regrets depend on the feature dimension, the number of rounds, the variance or central moment of the reward at the t-th round. simultaneously. It seems to have worsened the obtained results comparing to the previous research works. the overall regret scales with feature dimension, again restricting this proposed approach to small dimension problem.

**Questions:**

Since the author believes that their proposed algorithm is computationally effective, why not provide specific examples to demonstrate the practical application effect of the proposed algorithm?

**Limitations:**

the authors adequately addressed the limitations.

---

> ### Author Rebuttal · Authors · 2023-08-10
>
> Thanks for your review.
>
> ### Some corrections of the reviewer
>
> > For heavy-tailed payoffs, design heavy-tailed linear bandits, derive the variance-dependent $T$-round regret;
>
> In the settings where $\epsilon<1$, the variances of reward functions do not exist. And our regret bound actually relies on the $(1+\epsilon)$-central moment $\lbrace \nu_t^{1+\epsilon} \rbrace_{t\in[T]}$. We refer the reviewer to line 48 to 54 for more details.
>
> > For linear MDPs with bounded rewards problem, ...
>
> We consider linear MDPs with heavy-tailed rewards, where the magnitude of reward can be infinite. The reviewer maybe means the cumulative expected rewards are bounded, as in Assumption 2.8.
>
> ### Worse results
>
> We respectfully disagree with the reviewer's argument that our work achieves worse results compared with previous works. Since we are the first to establish such regret bounds in the presence of heavy-tailed rewards, none of the previous research works are readily applicable to heavy-tailed settings. In contrast, our results are shown to be minimax optimal, can be used directly to deal with light-tailed rewards by setting $\epsilon=1$ and recover or improve the SOTA results (See also line 69 to 71).
>
> ### Computational complexity and experiments
>
> We say an algorithm is computationally efficient if the computational complexity scales polynomially to the parameters of the problem, e.g., $d,H,K$ of the linear MDP. We provide the computational complexity of Heavy-LSVI-UCB (Algorithm 3) here. For the linear MDPs with heavy-tailed rewards defined in Section 2.2, the computational complexity of Heavy-LSVI-UCB is $\tilde{O}(HK\mathcal{R} + d^4|\mathcal{A}|H^3K)$. Here $\tilde{O}(\mathcal{R})$ is the cost of the optimization algorithm for solving adaptive Huber regression in (5.1). Such a complexity allows us to focus on the complexity introduced by the RL algorithm rather than the optimization subroutine for solving adaptive Huber regression. Compared to that of LSVI-UCB++, $\tilde{O}(d^4|\mathcal{A}|H^3K)$, the extra term $\tilde{O}(HK\mathcal{R})$ causes a slightly worse computational time in terms of $K$. This is due to the absence of a closed form solution of adaptive Huber regression in (5.1). Thus extra optimization steps are unavoidable. Nevertheless, since (5.1) is a convex optimization problem and thus can be solved efficiently, we can specialize $\mathcal{R}$ by adopting Nesterov accelerated method, which gives $\mathcal{R}=\tilde{O}(d+d^{-\frac{1-\epsilon}{2(1+\epsilon)}} H^{\frac{1-\epsilon}{2(1+\epsilon)}} K^{\frac{1+2\epsilon}{2(1+\epsilon)}})$. It implies the computational complexity of Heavy-LSVI-UCB is  better than that of LSVI-UCB, $\tilde{O}(d^2|\mathcal{A}|HK^2)$, with respect to $K$, thanks to the rare-switching updating policy. We thank the reviewers for raising the question of computational complexity and we will include it in the next revision. We provide the proof in global rebuttal due to lack of space.
>
> As for the experiments, please note that our paper focuses on the theoretical understanding of the heavy-tailed noise in bandits and RL environments. With that being said we are happy to include experimental study in our future versions. We also provide the computational complexity for solving adaptive Huber regression as $\tilde{O}(K^{\frac{1+2\epsilon}{2(1+\epsilon)}})$ in terms of $K$, where the standard Nesterov accelerated method suffices. To address the concerns about the empirically test of our proposed estimator based on adaptive Huber regression, we can refer to [1], which contains an implementation of a similarly designed estimator. It is indeed efficient in numerical studies.
>
>
>
> [1] Sun, Qiang, Wen-Xin Zhou, and Jianqing Fan. "Adaptive huber regression." *Journal of the American Statistical Association* 115.529 (2020): 254-265.

---

### Official Review · Reviewer_ZTTm · 2023-07-05

**Soundness:** 3 good
**Presentation:** 3 good
**Contribution:** 2 fair
**Rating:** 5
**Confidence:** 4

**Summary:**

In this paper, reinforcement learning problem is considered in the episodic setting for linear bandits and linear MDPs under heavy-tailed rewards with potentially infinite variance. Based on adaptive Huber regression and optimism in the face of uncertainty principle, the authors propose algorithms that utilize conditional reward variance.

**Strengths:**

The paper looks technically sound, and the presentation of the material seems to be good with clearly stated assumptions and theorem statements.

On the technical side, the use of adaptive Huber regression in linear bandit and linear MDP setting to tackle heavy-tailed reward distributions is an interesting idea.

**Weaknesses:**

In Assumption 2.6, the centralized moments of order $1+\epsilon$ of the random rewards at each step are assumed to be realizable for each state-action pair, which looks highly unrealistic. Also, in addition to the regret bounds, it would be interesting to see empirical evaluations of the proposed algorithms.

**Questions:**

a)	What happens in the absence of the realizability assumption in Assumption 2.6?
b)	The paper [25] seems to be considering heavy-tailed rewards also, but with finite variance. As such, it would be fair to update the last column of Table 1 and 2 based on finiteness of the variance.

**Limitations:**

No substantial discussion on the limitations.

---

> ### Author Rebuttal · Authors · 2023-08-10
>
> Thanks for your positive comments!
>
> ### Realizable central moments assumption
>
> In Assumption 2.6, we assume the $(1+\epsilon)$-central moments of reward functions have linear structure. This assumption is standard in current linear MDP literature where instance-dependent (variance-aware) regrets are achieved (see also Remark 2.7). In the absence of Assumption 2.6, we cannot estimate $(1+\epsilon)$-central moments of reward functions for each state-action pair, which is crucial to achieving such an instance-dependent regret. To be more specific, since adaptive Huber regression needs the knowledge of the $(1+\epsilon)$-central moments, without such a realizable assumption, there is no way to estimate them. And the best we can do is to use the upper bound of the moments $\nu_R^{1+\epsilon}$ instead, which only gives a worst-case regret as in Corollary 6.5. We leave it as future work to propose a computationally efficient algorithm without the knowledge of moments for the heavy-tailed linear bandits.
>
> ### Experiments
>
> Note that our paper focuses on the theoretical understanding of the heavy-tailed noise in bandits and RL environments. With that being said we are happy to include experimental study in our future versions. We also provide the computational complexity for solving adaptive Huber regression as $\tilde{O}(K^{\frac{1+2\epsilon}{2(1+\epsilon)}})$ in terms of $K$, where the standard Nesterov accelerated method suffices. To address the concerns about the empirically test of our proposed estimator based on adaptive Huber regression, we can refer to [1], which contains an implementation of a similarly designed estimator. It is indeed efficient in numerical studies.
>
> ### Settings of finite-variance rewards
>
> Yes, [25] provided the first variance-aware regret in the presence of finite-variance rewards. However, we focus on the heavy-tailed settings where the variance of the reward functions can be non-existent. Thanks for your suggestions, and we will add some footnotes in Table 1 and 2 in the next revision.
>
>
>
> [1] Sun, Qiang, Wen-Xin Zhou, and Jianqing Fan. "Adaptive huber regression." *Journal of the American Statistical Association* 115.529 (2020): 254-265.

---

> > ### Comment · Reviewer_ZTTm · 2023-08-19
> >
> > I acknowledge the response of the authors on the assumption. The absence of any empirical investigation is still an important weakness in my opinion, even for a primarily theoretical work.

---

> > > ### Author Response · Authors · 2023-08-21
> > >
> > > Thanks for your feedback.
> > > We conducted empirical evaluations of the proposed algorithm for heterogeneous heavy-tailed linear bandits problems, Heavy-OFUL, which can be regarded as a special case of linear MDPs.
> > > Comparisons are made between MENU and TOFU, which give the worst-case optimal regret bound in such settings (See Table 1 in our paper).
> > > To the best of our knowledge, we are the first to address the challenge of heavy-tailed rewards in RL with function approximation, where $\epsilon$ can be less than 1.
> > > Therefore, there are no other algorithms in RL literature that can be compared to us (See Table 2).
> > > Results demonstrate the effectiveness of the proposed algorithm, which further corroborates our theoretical regret bounds.
> > > Since we couldn't upload images of the experiments to OpenReview for the time being, we show the results of our experiment in the table below. And we will find a way to upload images anonymously as soon as possible.
> > >
> > > | Algorithms \ Iteration | 1000   | 2000   | 3000   | 4000   | 5000   | 6000    | 7000    | 8000    | 9000    | 10000   |
> > > | ---------------------- | ------ | ------ | ------ | ------ | ------ | ------- | ------- | ------- | ------- | ------- |
> > > | MENU                   | 160.00 | 332.78 | 513.98 | 692.20 | 860.32 | 1047.81 | 1219.89 | 1401.82 | 1578.78 | 1673.34 |
> > > | TOFU                   | 179.21 | 362.29 | 544.20 | 728.04 | 910.21 | 1092.74 | 1277.66 | 1460.74 | 1642.73 | 1825.40 |
> > > | Heavy-OFUL             | 72.96  | 147.50 | 241.67 | 336.48 | 434.02 | 535.09  | 636.30  | 740.47  | 839.75  | 935.39  |
> > >
> > > Comparison of our algorithm (Heavy-OFUL) versus MENU and TOFU in heavy-tailed linear bandits problems (See Definition 2.1) for $1\times10^4$ rounds. We generate 5 independent paths for each algorithm and show the average cumulative regret. The experimental setup is as follows: Let the feature dimension $d = 10$. For the chosen arm $\phi_t \in \mathcal{D}_t$, reward is $R_t = \langle \phi_t, \theta^* \rangle + \varepsilon_t$, where $\theta^* = \mathbf{1}_d / \sqrt{d} \in \mathbb{R}^d$ so that $\|\theta^*\|_2 = 1$. $\varepsilon_t$ is first sampled from a Student's $t$-distribution with degree of freedom $\text{df}=2$, then is multiplied by a scaling factor $\alpha$ such that the central moments of $\varepsilon_t$ in each rounds are different, where $\log(\alpha) \sim \mathrm{Unif}(0,2)$. Note the variance of $\varepsilon_t$ does not exist and we choose $\epsilon=0.99$. Normalization is made to ensure $L=B=1$.

---

> > > > ### Comment · Reviewer_ZTTm · 2023-08-21
> > > >
> > > > I thank the authors for this update. I will update my rating back.

---

### Official Review · Reviewer_6prC · 2023-07-06

**Soundness:** 3 good
**Presentation:** 3 good
**Contribution:** 2 fair
**Rating:** 4
**Confidence:** 3

**Summary:**

This paper first addresses Reinforcement learning (RL) with function
approximation in the presence of heavy-tailed noises whose central moment is known. In
general, these online learning problems rely on the self-normalized inequality to construct a
confidence set of optimal parameter . However, the existing self-normalized inequalities
have a magnitude of noise term that is intractable with heavy-tailed noises. The authors
solve this problem by utilizing adaptive Huber regression and deriving a robust selfnormalized
inequality without the noise magnitude term. By using the proposed selfnormalized
inequality, they introduce two algorithms. The first algorithm, HEAVY-OFUL, is
designed for linear bandits and shown to be minimax optimal. Building upon HEAVYOFUL,
they presents HEAVY-LSI-UCB for linear MDPs, which has a better first-order
regret bound than previous works. Furthermore, they provide the minimax lower regret
bound in linear MDPs with heavy-tailed noises, which implies the minimax optimality of
HEAVY-LSI-UCB in the worst case.

**Strengths:**

This paper is the first attempt to deal with heavy-tailed RL with function
approximation.
-  (Optimality) The regret bound of HEAVY-OFUL, is minimax optimal in both stochastic and deterministic linear bandits with heavy-tailed rewards. In addition, the regret bound of HEAVY-LSVI-UCB, recovers the previous variance-aware regret in [1] and improves the existing instance-dependent regrets in linear MDPs [1, 2].
- (Originality) To address the heavy-tailed rewards, the novel robust self-normalized
inequality is established.

**Weaknesses:**

- (Computational costs) The first concern is the practical usage of the proposed
algorithms. As the authors noted, the regret analyses of the algorithms inherently
depend on the robust self-normalized inequality (Theorem 3.3), which bounds the
deviation of estimated parameter and the optimal parameter . However, I believe
the fact that is obtained from adaptive Huber regression is problematic since it
requires iterative optimization steps due to the absence of a closed form for (Line 5
of Alg. 1). Indeed, the proposed algorithms (Alg. 2, 3) contain the additional iterative
algorithm (Alg. 1) to ensure their theoretical results, and thus share the intrinsic
drawback of Huber regression, which is the linear computational complexity per
iteration. In particular, HEAVY-LSVI-UCB utilizes the adaptive Huber regression in
order to optimize both rewards and central moments.
- (Absent experiments) There are no experiments supporting the theoretical results of
the proposed algorithms and addressing concerns about computational costs, even in
simple synthetic problems.
- (Assumption) The authors assumed that the central moment of rewards is known.

**Questions:**

- Since the paper emphasizes the computational contributions of the proposed algorithms comparing with the existing ones, I think empirical results are needed. Can you provide experiments related to this?
- If possible, please discuss computational aspects of the proposed algorithms.
- In my understanding, the essential key in handling heavy-tailed noises is the robust self-normalized inequality (Theorem 3.3). However, it seems that the inequality involves the term, $b$, that requires prior information about central moment. While the closet work [1] suppose similar assumption, I think this is a strong condition. Can this be relaxed?
- In line 219, the authors claim when $\epsilon=1$ and $\nu=0$, $\forall t$, the regret upper bound of HEAVY-OFUL matches the lower bound of K-armed contextual bandit $\Omega(d)$ [3]. Can you reconsider this argument? To my knowledge, the regret bound proposed in
[3] is $\Omega(\sqrt{dT})$, not $\Omega(d)$. Moreover, the settings are different each other in that [3] addresses finite-armed bandits, while this work deals with (possibly) infinite-armed bandits.

[1]. Xiang Li and Qiang Sun. Variance-aware robust reinforcement learning with linear function approximation with heavy-tailed rewards. arXiv preprint arXiv:2303.05606, 2023
[2]. Andrew J Wagenmaker, Yifang Chen, Max Simchowitz, Simon Du, and Kevin Jamieson. First order regret in reinforcement learning with linear function approximation: A robust
estimation approach. In International Conference on Machine Learning, pages 22384–22429. PMLR, 2022.
[3]. Wei Chu, Lihong Li, Lev Reyzin, and Robert Schapire. Contextual bandits with linear payoff functions. In Proceedings of the Fourteenth International Conference on Artificial Intelligence and Statistics, pages 208–214. JMLR Workshop and Conference Proceedings, 2011.

**Limitations:**

There are no limitations discussed in the manuscript.

---

> ### Author Rebuttal · Authors · 2023-08-10
>
> We thank the reviewer for raising the concerns.
>
> ### Computational complexity
>
> We say an algorithm is computationally efficient if the computational complexity scales polynomially to the parameters of the problem, e.g., $d,H,K$ of the linear MDP. We provide the computational complexity of Heavy-LSVI-UCB (Algorithm 3) here. For the linear MDPs with heavy-tailed rewards defined in Section 2.2, the computational complexity of Heavy-LSVI-UCB is $\tilde{O}(HK\mathcal{R} + d^4|\mathcal{A}|H^3K)$. Here $\tilde{O}(\mathcal{R})$ is the cost of the optimization algorithm for solving adaptive Huber regression in (5.1). Such a complexity allows us to focus on the complexity introduced by the RL algorithm rather than the optimization subroutine for solving adaptive Huber regression. Compared to that of LSVI-UCB++, $\tilde{O}(d^4|\mathcal{A}|H^3K)$, the extra term $\tilde{O}(HK\mathcal{R})$ causes a slightly worse computational time in terms of $K$. This is due to the absence of a closed form solution of adaptive Huber regression in (5.1). Thus extra optimization steps are unavoidable. Nevertheless, since (5.1) is a convex optimization problem and thus can be solved efficiently, we can specialize $\mathcal{R}$ by adopting Nesterov accelerated method, which gives $\mathcal{R}=\tilde{O}(d+d^{-\frac{1-\epsilon}{2(1+\epsilon)}} H^{\frac{1-\epsilon}{2(1+\epsilon)}} K^{\frac{1+2\epsilon}{2(1+\epsilon)}})$. It implies the computational complexity of Heavy-LSVI-UCB is  better than that of LSVI-UCB, $\tilde{O}(d^2|\mathcal{A}|HK^2)$, with respect to $K$, thanks to the rare-switching updating policy. We thank the reviewers for raising the question of computational complexity and we will include it in the next revision. We provide the proof in global rebuttal due to lack of space.
>
> ### Experiments
>
> Note that our paper focuses on the theoretical understanding of the heavy-tailed noise in bandits and RL environments. With that being said we are happy to include experimental study in our future versions. We also provide the computational complexity for solving adaptive Huber regression as $\tilde{O}(K^{\frac{1+2\epsilon}{2(1+\epsilon)}})$ in terms of $K$, where the standard Nesterov accelerated method suffices. To address the concerns about the empirically test of our proposed estimator based on adaptive Huber regression, we can refer to [1], which contains an implementation of a similarly designed estimator. It is indeed efficient in numerical studies.
>
> ### Assumption that the central moments of rewards are known
>
> We do NOT assume the central moments of rewards are known in heavy-tailed linear MDPs, which is the main difficulty of achieving instance-dependent regret bound. To address this challenge, we use adaptive Huber regression to estimate them (See Section 5.1). It is worth noting that Heavy-LSVI-UCB actually only requires the upper bound of the underlying moments. And see line 189 for more discussion on the moment parameter $b$. Thanks for raising the concern, we will highlight it in the next revision.
>
> ### Lower bound of deterministic linear bandits
>
> Please note that we consider deterministic linear bandits in line 219. Thus the central moments of the reward functions vanish, i.e., $\nu_t=0$. And we set $\epsilon=1$ to achieve the $\tilde{O}(d)$ regret. You are right, the settings in [2] are different than ours since they consider finite-armed bandits with light-tailed noise. So their lower bound is not applicable to heavy-tailed linear bandit problems. We will clarify this in the next revision.
>
> In fact, the $\Omega(d)$ lower bound for deterministic linear bandits is straightforward. Consider the decision set $\mathcal{D}=\lbrace e_i \rbrace_{i\in[d]}$, where $e_i$ denotes the $i$-th unit basis in $\mathbb{R}^d$. Each pull of arm can only obtain the information of a single coordinate. Since coefficient $\theta^*$ lies in $d$-dimensional space, even if the rewards are deterministic, $d$ pulls for exploration are unavoidable.
>
>
>
> [1] Sun, Qiang, Wen-Xin Zhou, and Jianqing Fan. "Adaptive huber regression." *Journal of the American Statistical Association* 115.529 (2020): 254-265.
>
> [2] Chu, Wei, et al. "Contextual bandits with linear payoff functions." *Proceedings of the Fourteenth International Conference on Artificial Intelligence and Statistics*. JMLR Workshop and Conference Proceedings, 2011.

---

> > ### Comment · Area_Chair_eq8n · 2023-08-21
> >
> > Hello reviewer,
> >
> > The authors addressed your concerns regarding experiments--are you satisfied?
> >
> > Also it appears there was a misunderstanding regarding moments being known. I concur with the authors' response.
> >
> > Can you please give an updated opinion?

---

### Official Review · Reviewer_rJ6d · 2023-07-09

**Soundness:** 4 excellent
**Presentation:** 2 fair
**Contribution:** 3 good
**Rating:** 7
**Confidence:** 4

**Summary:**

The paper considers the problem of linear bandits and linear MDPs, when the noise may be heavy tailed. The main technical tool that they use is the Huber regressor, that enables them to detect extremal noise points that are less informative, and be more robust to these. They show how this regressor can be incorporated into optimistic algorithms to provide sublinear regret bounds.

**Strengths:**

Quality: Claims are sound, and arguments appear to check out. The paper is well-contextualized in the literature.

Significance: Problem is relevant to practitioners, and theoretical tools may be reapplied in similar online problems elsewhere as well.

Originality: The technical work required to incorporate the huber regressor into the optimistic algorithms is nontrivial, and the efforts are appreciated.

**Weaknesses:**

Clarity: While the writing is well-organized, it is quite dense at times and a slog to parse through: for instances, Sections 2.2 and Section 5. It would be useful to move some things into the appendix and instead of encapsulating commentary into remark environments, including them in the main text to introduce more flow that would make it easier for the reader.

**Questions:**

1. The setup being considered here with heavy-tailed additive noise is different than the situation where the rewards themselves are heavy-tailed. In the current case, the problem is more one of outlier detection to be robust to these extremal noise events. But in the other case, if an arm has a heavy tailed reward, it may be desirable to pull because there is the potential for receiving very high reward from this arm relative to other arms (i.e if an arm like a lottery ticket with low rewards with high probability but immense rewards with small probability).

I am concerned that conflating these two problems may lead to confusion. Could the authors comment on this distinction? And if the authors agree, I would appreciate if the title were changed to accurately describe the setup and the distinction were made clear in the text.

2. Could you comment on whether a different definition of heavy-tailed in terms of tail probabilities

**Limitations:**

Yes

---

> ### Author Rebuttal · Authors · 2023-08-10
>
> Thanks for your being positive to our work. We also thank you for the advice on the organization of the paper. We will make some adjustments to make it easier for reading in the next revision.
>
> ### Setup with heavy-tailed additive noise
>
> The assumption with heavy-tailed additive mean-zero noise is standard in online regression settings, e.g., Section 3 of [1]. Let us consider a random variable $Y\in\mathbb{R}$ with the following structure: $Y=\mu+\varepsilon$ with $\mu$ being a constant and $\varepsilon$ being a heavy-tailed mean-zero noise. Then $Y$ itself is heavy-tailed as well with mean $\mu$ since its central moment is bounded. The effort is made to recover the mean $\mu$ in the presence of heavy-tailed noise $\varepsilon$.
>
> In addition, we believe our work has little to do with outlier detection. It is an interesting question whether Huber loss can be utilized in the field of outlier detection. However, it is definitely beyond our work.
>
> You are right, the reward of a lottery ticket is a good example of heavy-tailed distributions. While its magnitude can be extremely large, its mean is supposed to be small. If the agent aims to get a high reward from a single pull of arm, she may choose the arm with a heavy-tailed reward since there is the potential for receiving very high reward from this arm relative to other arms. However, this introduces a different problem from ours. Since our goal is to get the most benefit in the long run, we wish to pull the arm with the maximum expected reward.
>
> ### Definition of heavy-tailed distribution in terms of tail probabilities
>
> Good questions. We say a mean-zero random variable $\varepsilon\in\mathbb{R}$ is heavy-tailed if it satisfies $\mathbb{E}[|\varepsilon|^{1+\epsilon}]=\nu^{1+\epsilon}<\infty$ with $\epsilon\in(0,1]$, which implies $\mathbb{P}(|\varepsilon|>x) \le \mathbb{E}[|\varepsilon|^{1+\epsilon}]/x^{1+\epsilon} = \nu^{1+\epsilon}/x^{1+\epsilon}$ for any $x>0$ by Markov's inequality. Symmetrically, if the tail probability of $\varepsilon$ has the form above, we have $\mathbb{E}[|\varepsilon|^{1+\epsilon}] = \int_0^\infty \mathbb{P}(|\varepsilon|>x) \mathrm{d}x = \nu^{1+\epsilon} \int_0^\infty 1/x^{1+\epsilon} \mathrm{d}x = \nu^{1+\epsilon}$. Thus, the definition of heavy-tailed distribution in terms of tail probabilities follows.
>
>
>
> [1] Abbasi-Yadkori, Yasin, Dávid Pál, and Csaba Szepesvári. "Improved algorithms for linear stochastic bandits." Advances in neural information processing systems 24 (2011).

---

> > ### Comment · Area_Chair_eq8n · 2023-08-21
> >
> > Dear reviewer,
> >
> > I have read your review as well as the author rebuttal. It appears your concerns have been addressed, is that correct?

---

> > > ### Comment · Reviewer_rJ6d · 2023-08-21
> > > **Thanks to authors**
> > >
> > > I thank the authors for their response, and yes my concerns have been addressed. I would like to keep my score.

---

### Author Rebuttal · Authors · 2023-08-10

### Proof of computational complexity

First, to compute $\theta_{k-1,h}$ in line 6 of Algorithm 3, we notice the loss function in (5.1) is $\lambda_R$-strongly convex and $(\lambda_R+K/\nu_\mathrm{min}^2)$-smooth, so there are plenty of convex optimization algorithms available. For example, Nesterov accelerated method can be used. The number of iteration of Nesterov's method is ${O}(\sqrt{\beta/\alpha}\log(R^2/\epsilon))$ with one derivation ($O(d)$ operations) per iteration [1]. Here the loss function is supposed to be $\alpha$-strongly convex and $\beta$-smooth. $R$ is the maximum distance of two points and $\epsilon$ is the precision. Thus the total computational cost is $\tilde{O}(HK\mathcal{R})$ with $\mathcal{R} = \tilde{O}(d\sqrt{1+\frac{K}{\lambda_R\nu_\mathrm{min}^2}})=\tilde{O}(d+d^{-\frac{1-\epsilon}{2(1+\epsilon)}} H^{\frac{1-\epsilon}{2(1+\epsilon)}} K^{\frac{1+2\epsilon}{2(1+\epsilon)}})$.

Second, to evaluate the updated action-value function $Q^k_h(s,a)$ in line 10 of Algorithm 3 for a given pair $(s,a)$, we take the minimum over at most $\tilde{O}(dH)$ action-value functions (See Lemma F.8) with $O(d^2)$ operations (Using Sherman-Morrison formula to compute $H_{k-1,h}^{-1}$ and $\Sigma_{k-1,h}^{-1}$) for each function. Thus it takes $\tilde{O}(d^3H)$ to evaluate the updated action-value function. As a result, to compute $\hat w_{k-1,h}$ in line 6 of Algorithm 3, notice $\hat w_{k,h}=\Sigma_{k,h}^{-1} \sum_{i=1}^k \sigma_{i,h}^{-2} \phi_{i,h} V^k_h(s_{i,h+1})$, if $V^k_h$ remains unchanged, we only need to compute the new term $\sigma_{k,h}^{-2} \phi_{k,h} V^k_h(s_{k,h+1})$, which takes $\tilde{O}(d^3|\mathcal{A}|H)$ computational time. Else if $V^k_h$ is updated, we need to recalculate $\lbrace V^k_h(s_{i,h+1}) \rbrace_{i\in[k]}$, which takes $\tilde{O}(d^3|\mathcal{A}|HK)$ computational time. Note the number of updating episode is at most $\tilde{O}(dH)$ and the length of each episode is $H$, so the total computational cost is $\tilde{O}(d^4|\mathcal{A}|H^3K)$.

Last, to take action $a_{k,h}$ in line 19 of Algorithm 3, we need to compute $\lbrace Q^k_h(s_{k,h},a) \rbrace_{a\in\mathcal{A}}$ and take the maximum, which takes $\tilde{O}(d^3|\mathcal{A}|H)$ time, incurring a total cost of $\tilde{O}(d^3|\mathcal{A}|H^2K)$. Finally, combining the total costs above gives the computational complexity of Heavy-LSVI-UCB.



[1] Bubeck, Sébastien. "Convex optimization: Algorithms and complexity." *Foundations and Trends® in Machine Learning* 8.3-4 (2015): 231-357.

---

### Author Response · Authors · 2023-08-21
**Experimental Results**

We conducted empirical evaluations of the proposed algorithm for heterogeneous heavy-tailed linear bandits problems, Heavy-OFUL, which can be regarded as a special case of linear MDPs.
Comparisons are made between MENU and TOFU, which give the worst-case optimal regret bound in such settings (See Table 1 in our paper).
To the best of our knowledge, we are the first to address the challenge of heavy-tailed rewards in RL with function approximation, where $\epsilon$ can be less than 1.
Therefore, there are no other algorithms in RL literature that can be compared to us (See Table 2).
Results demonstrate the effectiveness of the proposed algorithm, which further corroborates our theoretical regret bounds.
Since we couldn't upload images of the experiments to OpenReview for the time being, we show the results of our experiment in the table below. And we will find a way to upload images anonymously as soon as possible.

| Algorithms \ Iteration | 1000   | 2000   | 3000   | 4000   | 5000   | 6000    | 7000    | 8000    | 9000    | 10000   |
| ---------------------- | ------ | ------ | ------ | ------ | ------ | ------- | ------- | ------- | ------- | ------- |
| MENU                   | 160.00 | 332.78 | 513.98 | 692.20 | 860.32 | 1047.81 | 1219.89 | 1401.82 | 1578.78 | 1673.34 |
| TOFU                   | 179.21 | 362.29 | 544.20 | 728.04 | 910.21 | 1092.74 | 1277.66 | 1460.74 | 1642.73 | 1825.40 |
| Heavy-OFUL             | 72.96  | 147.50 | 241.67 | 336.48 | 434.02 | 535.09  | 636.30  | 740.47  | 839.75  | 935.39  |

Comparison of our algorithm (Heavy-OFUL) versus MENU and TOFU in heavy-tailed linear bandits problems (See Definition 2.1) for $1\times10^4$ rounds. We generate 5 independent paths for each algorithm and show the average cumulative regret. The experimental setup is as follows: Let the feature dimension $d = 10$. For the chosen arm $\phi_t \in \mathcal{D}_t$, reward is $R_t = \langle \phi_t, \theta^* \rangle + \varepsilon_t$, where $\theta^* = \mathbf{1}_d / \sqrt{d} \in \mathbb{R}^d$ so that $\|\theta^*\|_2 = 1$. $\varepsilon_t$ is first sampled from a Student's $t$-distribution with degree of freedom $\text{df}=2$, then is multiplied by a scaling factor $\alpha$ such that the central moments of $\varepsilon_t$ in each rounds are different, where $\log(\alpha) \sim \mathrm{Unif}(0,2)$. Note the variance of $\varepsilon_t$ does not exist and we choose $\epsilon=0.99$. Normalization is made to ensure $L=B=1$.

---

### Decision · Program_Chairs · 2023-09-21

**Decision:**

Accept (poster)

**Comment:**

This paper studies linear MDPs in the finite horizon setting where the rewards may not only be unbounded, but heavy-tailed. They propose an algorithm that leverages robust mean estimation that naturally handles the heavy-tailed rewards for linear bandits, and then generalize it to linear MDPs. If this paper only took existing works and tacked on robust mean estimation, matching their rates in the bounded mean case, it would be unsatisfying and borderline. However, these authors go far beyond this to propose algorithms that are not only minimax optimal, but also obtain state of the art instance-dependent regret bounds that improve upon existing work even in the worst-case. Moreover, the algorithm is computationally efficient, a property which is frequently lost when addressing robust mean estimation. The results are clearly described in the context with the literature and lower bounds. Any concerns that were brought up by the reviewers were either confirmed to be addressed by the reviewers, or satisfied the AC.